# Relationship between Bone Quality, Egg Production and Eggshell Quality in Laying Hens at the End of an Extended Production Cycle (105 Weeks)

**DOI:** 10.3390/ani11030623

**Published:** 2021-02-26

**Authors:** Clara Alfonso-Carrillo, Cristina Benavides-Reyes, Jon de los Mozos, Nazaret Dominguez-Gasca, Estefanía Sanchez-Rodríguez, Ana Isabel Garcia-Ruiz, Alejandro B. Rodriguez-Navarro

**Affiliations:** 1Trouw Nutrition R&D, Ctra. CM 4004, km 10.5, Casarrubios del Monte, 45950 Toledo, Spain; c.alfonso@trouwnutrition.com (C.A.-C.); j.delosmozos@trouwnutrition.com (J.d.l.M.); 2Departamento de Mineralogía y Petrología, Universidad de Granada, 18002 Granada, Spain; crisbr@ugr.es (C.B.-R.); nadoga@ugr.es (N.D.-G.); estefaniasr@outlook.com (E.S.-R.)

**Keywords:** laying hens, bone quality, osteoporosis, aging, eggshell quality

## Abstract

**Simple Summary:**

Nowadays, the industry aims to extend egg laying until hens are 100 weeks old or longer (from 65 to 70 weeks currently) to make egg production more sustainable. However, intensive egg production challenges hen health and particularly bone metabolism as eggshell formation mobilizes large amounts of calcium from the skeleton, inducing a severe form of osteoporosis and bone fractures. In this study, bone quality was compared in old hens (100 weeks) having different egg production and eggshell quality. Hens with a high egg production and good eggshell quality had poorer bone quality than hens with lower production and/or bad eggshell quality. This is due to a greater calcium mobilization capacity that decreases bone mineral reserves in the form of medullary bone. However, no clear relationship could be found between measured bone quality and egg production/eggshell quality parameters, suggesting that both bone and egg quality can be improved independently through genetic selection and optimum nutrition.

**Abstract:**

(1) Background: Nowadays the industry aims to improve lay persistency for extended cycles (100 weeks or longer) to make egg production more sustainable. However, intensive egg production challenges hen health, inducing severe osteoporosis and the incidence of bone fractures. In this study, the relationship between bone quality and egg production, and/or eggshell quality, was evaluated at the end of an extended laying cycle of 100 weeks, comparing groups of hens with different production and eggshell quality parameters; (2) Methods: Quality parameters of egg (as weight, egg white height), eggshell (as thickness, weight, breaking strength, elasticity and microstructure) and tibiae bone (weight, diameter, cortical thickness, ash weight, breaking strength, medullary bone) were determined; (3) Results: Hens from groups with a high egg production and good eggshell quality have poorer bone quality (lower ash weight and lesser amount of medullary bone). However, Pearson’s correlation analysis shows no clear relationship between bone and egg/eggshell parameters. (4) Conclusions: Bone and egg production/eggshell quality are independent and can be improved separately. Medullary bone has an important contribution to bone mechanical properties, being important to accumulate enough bone medullary bone early in life to maintain skeletal integrity and eggshell quality in old hens.

## 1. Introduction

Egg laying is highly demanding for the organism and causes a general deterioration of the hen health during the intensive egg laying period [1,2]. Particularly, there is a deterioration of skeletal integrity with hen age as large amounts of calcium (about 2 g daily) are exported for eggshell formation coming in part in the diet and from the skeleton, mainly from medullary bone [3,4,5]. Medullary bone starts forming when hens reach sexual maturity, about two weeks before they start laying. During this period, the hen skeletal weight increases by 20%. Medullary bone mineral is highly reactive and has a rapid turnover rate so that it supplies the calcium needed during the night when eggshell is being mineralized and dietary calcium in the gut may be exhausted [5]. However, during the intensive egg laying period, medullary bone is formed at the expense of cortical bone resulting in a progressive loss of structural bone especially if the calcium supply in the diet is not adequate [2,3,4,6]. The development of avian osteoporosis, and the high incidence of associated bone fractures, particularly at the end of the laying cycle, is one of the most relevant welfare problems facing the egg industry today [4,7,8,9,10]. Moreover, birds affected by bone fractures have a decreased egg production and increased food intake and mortality [11,12,13].

On the other hand, there is also a decrease of eggshell quality during the production cycle. The number of cracked and damaged eggs can increase up to 20–30% at the end of the laying cycle (65 to 70 weeks), being one of the main reasons to limit the production to that age. The decrease in eggshell quality in older hens is due in part to a reduction of calcium absorption by the intestine and an increase in the size of eggs resulting in a decreased shell thickness and shell weight percentage [14,15]. There are also changes in the eggshell structure with hen age that further decrease eggshell strength [16,17]. Poor eggshell quality is a major concern for food safety as eggs with a damaged eggshells are more easily contaminated with bacteria [18].

Hens that do not lay eggs have stronger, more mineralized bones with larger amounts of medullary bone [9,19]. Thus, it is clear that egg laying and high calcium requirements for eggshell formation make hens more susceptible to develop skeletal problems such as osteoporosis [3,13,20]. However, it is not clear that a negative relationship exists between egg production and bone quality or that selection for high egg production is responsible for skeletal weakness [21,22,23]. In fact, it is possible to improve hens’ bone quality without reducing egg production [4]. However, hens with higher bone quality might lay eggs with poorer eggshell quality [10,20]. In contrast, hens with low bone quality could have a more active bone resorption to supply more calcium for eggshell formation, resulting in better eggshell quality.

Nowadays, the industry aims to improve the laying persistency until hens are 100 weeks in a single cycle without molting to make egg production sustainable. However, this could exacerbate skeletal and egg quality problems at the end of laying cycle [1,2]. It is important to understand the relationship between egg production and eggshell quality with skeletal integrity, especially in extended production cycles for which information is lacking.

The aim of the present work is to study bone quality and its relationship with egg production and/or eggshell quality in laying hens at the end of an extended production cycle until hens are 100 weeks old. This work used complementary analytical techniques such as scanning electron microscopy (SEM), infrared spectroscopy (FTIR), and thermogravimetry (TGA) to characterize tibiae bone quality from groups of hens that have been classified according to their egg production (high or low) and eggshell quality (good or bad). These techniques provide information on important bone characteristics such as bone structural organization at different levels, porosity, and chemical composition, which have important contributions to bone properties (i.e., mechanical properties) and are useful to define bone quality [24]. The information gathered could help develop new strategies (based on selection and/or nutrition) to maintain bone quality, egg laying performance and good eggshell quality in older hens.

## 2. Materials and Methods

This study was conducted at the Trouw Nutrition Poultry Research Facility (Ctra. CM-4004, KM. 10.5, 45950 Casarrubios del Monte, Spain) using 144 Brown Nick (H&N) laying hens housed during the rearing and laying period in enriched commercial cages. All animals were fed ad libitum with the same diets, which were formulated following the nutrient requirements of different production phases of CVB recommendations. A temperature of 20 to 22 °C and a 16/8 h light/dark period were used. When hens reached 98 weeks, they were moved to individual cages (0.3 m width × 0.5 m length × 0.32 m high) to monitor their egg production (lay percentage) and egg quality (cracked eggs) individually during the experimental phase (a five-week period). A two-week period was used as adaptation to the new housing system and afterwards, at the end of the 5-week experimental period, 100 hens were selected according to their egg production (high, HP; or low, LP) and eggshell quality (Good, GQ; or Bad, BQ) and assigned to four different groups with 25 birds per group (Table 1). 

At the end of this period, all of the birds were slaughtered by cervical dislocation according to the guidelines of the Ethics Committee of Poultry Research Centre of Trouw Nutrition for the humane care and use of animals in research. Animals were dissected to excise the ovaries and uterus and to collect the tibiae. Bones were kept in a freezer at −20 °C until analyzed.

### 2.1. Egg Properties

All eggs produced during the last two weeks (n = 932) of the experimental period were analyzed in detail for egg quality traits such as egg weight, egg white height (HU), eggshell thickness (at the equator), eggshell weight and eggshell breaking strength, and elasticity, were measured. For the eggshell mechanical properties, a Stable Micro Systems TA XT plus 100 testing machine was used. Additionally, the eggshell microstructure was analyzed in selected samples by optical and electron microscopy. To do this, eggshells were embedded in epoxy resin to prepare double-polished thin-sections (<30 μm) for observation using a polarized light optical microscope (Nikon LZM 1000, Japan). Also, intact eggshell samples were observed in cross-section by scanning electron microscopy (SEM) (Hitachi S-510, Hitachi Ltd., Tokyo, Japan).

### 2.2. Bone Properties

Both tibias from all animals were collected at the end of the experiment. Tibias from the left leg were used to determine the main morphological properties such as weight and diameter, mechanical properties (e.g., tibia breaking strength and elasticity), bone cortical thickness and the total mineral content as ash weight of whole tibia. Tibia breaking strength and elasticity were determined by a three-point bending test using a material testing machine (Stable Micro Systems TA. XT plus 100) with a 100 kg loading cell and speed of 3 mm/s. The ash weight of whole tibia was determined by heating in a Carbolite oven (model RWF 1100) at 550 °C for 12 h.

Tibias from the right leg were used in the scanning electron microscopy (SEM) study and mineral composition in cortical and medullary bone assessment. For the SEM study, five samples for each group were selected to prepare polished bone cross-sections of the tibiae mid-diaphysis. Bone samples were embedded in Epothin epoxy resin (Buehler), cut, polished, coated with carbon and observed with a SEM using backscattering electron mode (BSE) at an accelerating voltage of 25 keV.

The cortical and medullary bone collected from the tibiae mid-diaphysis were manually separated using a scalpel and homogenized by grinding in a stainless-steel ball mill (Mini Mill Pulverisette 23, Fritsch GmbH, Idar-Oberstein, Germany). The absolute water, organic matter, and mineral content in the cortical and medullary bone samples were determined by thermogravimetry (TGA). For these analyses, about 25 mg of powered cortical and medullary bone samples were introduced into silver capsules 8 × 5 mm (Elemental Microanalysis Ltd., Devon, UK) and heated in a laboratory chamber furnace ELF 11/14 (Carbolite^®^, Barloworld Scientific Ltd., Burlington, NJ, USA). A heating rate of 10 °C/min in air from room temperature to 900 °C was used for registering the TGA curves. Samples were weighed in a microbalance Sartorius M2P (ISO 9001) (Sartorius, Göttingen, Germany) at room temperature, 200, 600 and 900 °C to obtain the percentage of water, organic matter, and mineral.

The chemical compositions of bone tissues (cortical and medullary bone) were also analyzed by infrared spectroscopy using a FTIR spectrometer JASCO 6200 equipped with a diamond attenuated total reflection (ATR) accessory (ATR Pro ONE, JASCO) following the protocol fully described elsewhere [24] and the relative amount of mineral to organic matrix (degree of mineralization; PO4/Amide I) determined as the ratio of the main phosphate (v1, v3 PO4; 900–1200 cm^−1^) to Amide I (1590–1710 cm^−1^) band area ratio were calculated. Additionally, the eggshell cuticle was assessed by ATR-FTIR.

### 2.3. Statistical Analyses 

The basic descriptive statistics (mean and standard deviation) of each parameter were calculated. Two-way ANOVA was used to compare egg and bone properties between hens from groups with different egg production and eggshell quality. The model included the fixed effects as the production level, eggshell quality, and their interactions. Also, differences between groups were determined with Tukey’s test. Pearson’s correlation analysis and multivariate linear regression models were used to study the relationships between the analyzed variables and bone mechanical properties. All statistical analyses were performed using SAS (SAS Inst.) and Origin Pro (Microcal) software packages. A *p* < 0.05 value was considered significant.

## 3. Results

### 3.1. Animal Production

Table 2 summarizes the main parameters determined in hens from experimental groups with different egg production and egg quality. ANOVA tests show that hens from high-production groups had a larger body weight and higher feed intake than low-production ones. Egg production varied from 75% for high-production (HP) groups to 55% for low-production (LP) groups, these values being significantly different (*p* < 0.001). Egg production of LPBQ was lower than LPGQ (56% vs. 60%; *p* < 0.05). There were no significant differences in ovaries and uterus weight between groups, though hens with high production and good eggshells (HPGQ) had a notably larger uterus.

### 3.2. Egg Properties

Basic parameters related to egg quality are also summarized in Table 2. The egg weight is large (about 67 g) and eggshell quality parameters such as eggshell thickness (0.36 to 0.33 mm), shell percentage (8.9–7.9%) are relatively low, as would be expected for old hens. There were no significant differences between groups in egg weight or egg white height. However, there were notable differences in parameters related to eggshell quality. Specifically, the percentage of cracked eggs was much lower in good quality (GQ) groups (about 6%) than in bad quality (BQ) groups (about 21%). Other eggshell quality traits such eggshell thickness, eggshell weight, shell percentage, elasticity and breaking strength were significantly larger in GQ than in BQ groups. No significant differences were observed among groups for the amount of cuticle. Examination of the ultra- and microstructure of eggshells by optical and electron microscopy showed some notable differences between the GQ and BQ groups (Figure 1). Eggshells from the BQ groups generally exhibited a number of structural defects such as late fusion of the mammillary layer, low mammillary density, and/or poor membrane attachment (Figure 1A) that were not observed so often in the GQ groups (Figure 1B).

#### Correlations between Egg Properties

Pearson’s correlation analyses for egg properties, reported in Table 3, show some interesting relationships among measured parameters. For instance, body weight was positively correlated to uterus weight, egg weight, shell weight and lay percentage. The uterus weight was positively correlated to egg weight, shell weight and lay percentage (*p* < 0.001). The eggshell breaking strength was highly and positively correlated with shell weight, shell thickness and even more strongly to shell percentage (R = 0.831; *p* < 0.001).

### 3.3. Bone Properties

Figure 2 displays representative images of the cross-section of tibia bones from laying hens of the different experimental groups, showing the distribution of cortical and medullary bone. Cortical bone generally shows the presence of resorption centers (RC) where it is being partially replaced by medullary bone indicating the development of osteoporosis. Medullary bone is formed by small isolated trabeculae distributed in the proximity of the endosteal surface of cortical bone. After examining bone from each experimental group, clear differences could not be observed between the porosity of cortical bone (number and size of resorption centers) and the amount of medullary bone among the four groups due to the large variability. Nevertheless, examined bones from high-production groups seem to have a reduced amount of medullary bone than the low-production groups. 

Table 2 summarizes the main tibiae bone characteristics determined for each experimental group. Significant differences in tibiae weight (*p* = 0.006) and bone breaking strength (*p* = 0.040) were found between eggshell quality groups. Hens with bad eggshell quality had heavier and stronger bones than hens with good eggshell quality. Bones from the LPBQ group had the greatest breaking strength and mineral content (ash weight). No effect on egg production or eggshell quality was observed for the other variables (bone diameter, cortical bone thickness and elasticity).

No differences were observed on the mineral content of cortical and medullary bone between groups, though the LPBQ group had the largest mineral content in the medullary bone which is in agreement with their having the highest ash weight. All in all, it seems that hens from the low-production and/or bad-eggshell-quality groups mobilize less calcium; as a result, their bones retain more medullary bone.

#### Correlations between Bone Properties

Pearson’s correlation analyses for bone properties, reported in Table 4, show that there were some notable and interesting relationships among measured bone parameters. Body weight is positively correlated with most bone properties, such as bone weight, bone diameter, cortical thickness, ash weight and bone breaking strength. On the other hand, bone diameter is negatively correlated to cortical thickness and the degree of mineralization of cortical bone (Mineral_CB). During bone growth, the outer and inner diameter of bone increase due to resorption at the inner surface (endosteal surface) and apposition of new bone at the outer surface (periosteal surface). If cortical thickness decreases, it means that endosteal resorption is more intensive than periosteal apposition. The whole bone ash weight was positively correlated to cortical thickness and even more strongly to the amount of medullary bone mineral (%Mineral_MB), indicating that the variation in mineral content is mainly due to medullary bone and, to a lesser degree, cortical bone.

Bone breaking strength was correlated most strongly with whole bone ash weight (R = 0.751; *p* < 0.001). Additionally, the bone breaking strength was correlated with the cortical thickness (R = 0.327; *p* = 0.001) and more strongly to the amount of medullary bone (R = 0.424; *p* < 0.001), indicating that not only cortical but also medullary bone contributes to the mechanical properties of bone. If cortical and medullary bone parameters (Cortical thickness, Mineral_CB, Mineral_MB) are put together in a multivariate linear model, the correlation with breaking strength increases but is still much lower than the tibia ash weight alone. Note that other bone properties, except whole tibia ash weight, were determined locally at the tibiae midshaft, which is more mineralized than other regions of bone (distal parts). More representative measurements of global properties, such as the ash weight of whole tibiae, are needed to explain the bone breaking strength, which is also measured in whole tibiae. 

### 3.4. Relationship between Bone Properties and Egg Parameters

To better understand the relationship between bone quality and egg production/quality, Pearson’s correlation analysis between tibiae bone and egg properties was carried out (Table 5). Most of the bone and egg-related parameters considered did not hold any significant relationship among them. There was a negative correlation between uterus weight and the mineral content in cortical bone. There was also a positive correlation between egg weight and tibiae diameter. All in all, this study shows that the relationship between bone and egg production/quality is rather weak or not obvious.

## 4. Discussion

The present work aimed to study in detail the relationship between bone characteristics and egg production and/or eggshell quality in 100-week-old hens at the end of an extended production cycle. This study shows that hens at the end of this long cycle produce eggs with poor eggshell quality and also have poor bone quality with notable signs of osteoporosis irrespective of the egg production levels. Hens with a high production and good eggshell quality are heavier and have a slightly larger uterus (shell gland) than the other groups with lower production and/or poorer eggshell quality. Hens with a high production and good eggshell quality have a greater capacity to mobilize the calcium needed for eggshell formation and retain a lesser amount of medullary bone. On the other hand, hens from the low-production and/or poor-eggshell-quality groups have a greater amount of bone mineral (i.e., higher ash weight and greater amount of medullary bone) which might indicate that they have a decreased capacity to mobilize calcium and therefore accumulate a larger amount of medullary bone.

The skeleton of birds contributes with a significant amount of calcium for eggshell formation (up to 40%) [2,5]. Bone mineral resorption, mainly from medullary bone, due to high calcium requirements for eggshell formation is the main cause of osteoporosis in laying hens, a condition that develops with age and/or poor nutrition (low calcium diets) [1,3,4,6]. Thus, it is reasonable to assume that a negative relationship could exist between egg production and skeletal integrity or bone quality [3,10,20]. Kim et al. (2005) [25] observed a strong negative correlation between eggshell quality (i.e., shell weight) and bone mineralization (i.e., bone ash weight). Whitehead (2004) [3] also shows that there is a negative relationship between bone quality and egg production; however, when low-production hens are excluded (<250 egg per year), this relationship is almost lost. This author argues that hens with low egg production have been out of lay for some periods and could have built up new bone. Also, hens that have been selected for high bone quality have slightly poorer eggshell quality [4,10]. Thus, it seems that laying hens that are more susceptible to osteoporosis are able to mobilize more calcium and lay eggs with better eggshell quality [4,20,25,26]. On the other hand, chickens that do not lay eggs (males or non-ovulating hens) have stronger, more mineralized bones or larger amounts of medullary bone [9,19]. Thus, it is clear that egg laying and high calcium requirements for eggshell formation makes hens more susceptible to develop skeletal problems such as osteoporosis [3,13,20]. However, other authors suggest that there is no evidence that a relationship exists between bone quality and egg production and eggshell quality, or that selection for high egg production is responsible for skeletal weakness [21,22,23]. In fact, in this study, no evident relationships were found between the examined characteristics related to bone quality and egg production or eggshell quality. These results are in agreement with recent studies showing very weak or no evidence of phenotypic or genotypic relationships between bone quality and egg production or egg quality [22,23]. Thus, it seems that bone and egg characteristics are mostly independent and can be improved separately. On the other hand, studies monitoring the age at the first egg show a strong association between this parameter and bone quality and the incidence of (keel) bone fractures [23]. Selection of hens for early sexual maturity and peak egg production might have prevented them from accumulating enough medullary bone to maintain skeletal integrity throughout the intensive egg production period [2,23]. It might also have not allowed for an adequate development of the oviduct for egg laying.

Even though no clear relationship between bone and egg parameters was observed in the current study, other correlations among egg and bone parameters alone were found. Specifically, there was a significant correlation between uterus weight with egg weight, shell weight and lay percentage. This is not surprising since the uterus (shell gland) is the main organ involved in eggshell deposition and a well-developed uterus is needed to sustain egg and eggshell formation.

Body weight was positively correlated with bone weight, bone diameter, cortical thickness and breaking strength. This seems logical as bones adapt to mechanical loads, stimulating bone formation [4,24]. The bone diameter was negatively correlated to cortical thickness due to a strong endosteal resorption activity associated with medullary bone mobilization for eggshell formation. Most of the variation in bone mineral content was due to medullary bone and, to a lesser degree, cortical bone mineralization. Additionally, it was found that not only cortical bone, but also medullary bone has an important contribution to bone mechanical properties. Thus, it is very important to accumulate and maintain enough medullary bone for skeletal structural integrity and bone quality, in agreement with other studies [23,24,27]. Still, the best proxy for bone strength was whole bone ash weight which is, again, highly correlated to medullary bone content. 

Many relevant bone and egg characteristics are known to have an important genetic component that can be used in genetically assisted selection programs aimed at improving bone and eggshell quality [1,27,28,29]. Also, adequate hen nutrition is crucial to achieve the genetic potential of modern layers while maintaining overall laying hen health [2,30,31,32]. For instance, as calcium absorption decreases with age, Ca% in the diet needs to be increased at the end of lay [2,33]. Also, supplying calcium as coarse calcium carbonate particles improves eggshell quality and reduces calcium mobilization from medullary bone, maintaining skeletal integrity [4,33].

## 5. Conclusions

In conclusion, maintaining a high production of eggs with good eggshell quality in an extended production cycle (until hens are over 100 weeks) challenges hen health in general and bone metabolism in particular. Hen skeletal integrity and eggshell quality deteriorates with age. This study shows that laying hens with low egg production and bad eggshell quality have better bone quality (a higher breaking strength and amount of medullary bone) due to decreased bone mineral resorption. However, no clear relationship between bone quality and egg production/eggshell quality parameters could be found, suggesting that bone and egg quality traits can be improved independently through genetic selection and optimum nutrition. Hens need to build up enough medullary bone early in life to maintain calcium reserves and skeletal integrity during longer egg production periods. On the other hand, medullary bone has a significant contribution to bone mechanical properties. Thus, allowing hens to accumulate larger amounts of medullary bone (for instance, delaying age at first egg; optimizing diet during prelay) could reduce osteoporosis and bone fractures. Additionally, hens need to develop and maintain a healthy and functional oviduct to sustain such a high egg production with good eggshell quality. 

## Figures and Tables

**Figure 1 animals-11-00623-f001:**
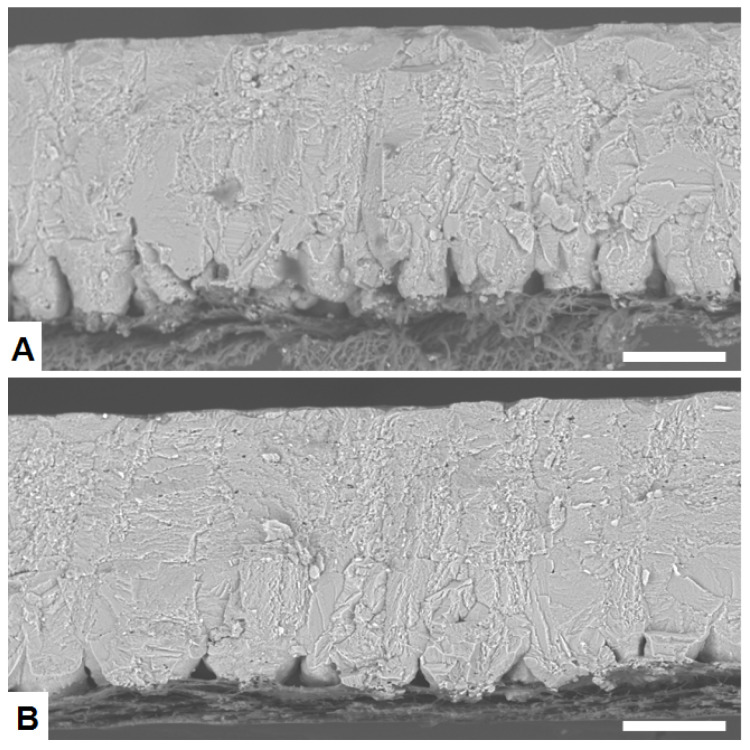
Representative scanning electron microscopy (backscattering electron mode (BSE-SEM)) images of the eggshell ultrastructure from: (**A**) Bad-eggshell-quality group showing late fusion of mammillary knobs; (**B**) Good-eggshell-quality group showing a denser structure. Scale bar: 100 µm.

**Figure 2 animals-11-00623-f002:**
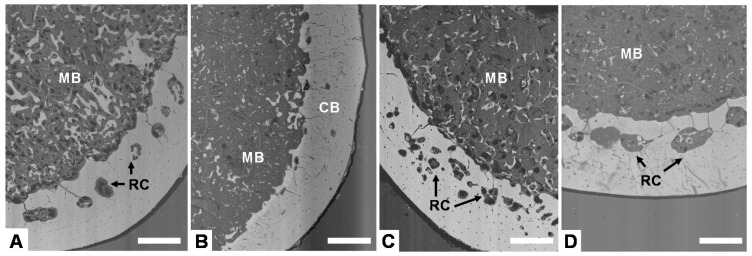
Backscattering electron (BSE-SEM) images of tibia cross-section at mid-shaft from hens of different groups: (**A**) High production, good quality; (**B**) High production, bad quality; (**C**) Low production, good quality; (**D**) Low production, bad quality. CB: cortical bone. MB: Medullary bone. RC: resorption center. Scale bar: 500 µm.

**Table 1 animals-11-00623-t001:** Experimental groups of laying hens with different egg production and egg quality.

Group	N	Egg Production	Eggshell Quality
Group HPGQ	25	>70%	<10% cracked eggs
Group HPBQ	25	>70%	>10% cracked eggs
Group LPGQ	25	<70% (30% minimum)	<10% cracked eggs
Group LPBQ	25	<70% (30% minimum)	>10% cracked eggs

**Table 2 animals-11-00623-t002:** Hen-, egg-, and tibia-related parameters. HP: High production; LP: Low Production; GQ: Good Quality; BQ: Bad Quality; HU: Haugh unit; CB: Cortical Bone; MB: Medullary Bone. Significant different are highlighted in bold. Differences between groups (Tukey´s tests) indicated by a, b.

	HP		LP		Probability (P)		
	GQ	BQ	GQ	BQ	Production	Quality	Interaction
Body weight (g)	1938 ± 175 ^a^	1858 ± 168 ^a,b^	1777 ± 230 ^b^	1847 ± 202 ^a,b^	**0.030**	0.896	0.059
Feed intake (g/d)	114.31 ± 11.64 ^a^	108.93 ± 20.97 ^a,b^	104.88 ± 17.47 ^a,b^	101.62 ± 15.70 ^b^	**0.013**	0.654	0.647
Ovaries weight (g)	49.85 ± 8.89	49.32 ± 10.99	49.00 ± 13.67	48.37 ± 9.79	0.688	0.795	0.983
Uterus weight (g)	77.56 ± 10.59	72.03 ± 14.92	71.90 ± 13.31	71.11 ± 8.50	0.174	0.187	0.321
Lay percentage (%)	75.03 ± 11.13 ^a^	75.66 ± 11.36 ^a^	60.02 ± 11.37 ^b^	55.58 ± 11.61	**<0.001**	0.111	**0.034**
Cracked eggs (%)	6.49 ± 16.63	20.72 ± 16.97	5.29 ± 16.98	20.56 ± 17.34	0.703	**<0.001**	0.769
Egg weight (g)	67.23 ± 5.58	66.88 ± 4.35	67.00 ± 3.90	67.58 ± 5.02	0.728	0.863	0.499
Egg white height (HU)	71.72 ± 8.48	72.24 ± 12.37	71.97 ± 10.00	71.98 ± 11.19	0.822	0.996	0.678
Eggshell weight (g)	6.04 ± 0.48 ^a^	5.32 ± 0.67 ^b^	5.91 ± 0.56 ^a^	5.32 ± 0.71 ^b^	0.454	**<0.001**	0.447
Eggshell thickness (mm)	0.36 ± 0.02 ^a^	0.33 ± 0.03 ^b^	0.36 ± 0.03 ^a^	0.33 ± 0.04 ^b^	0.684	**<0.001**	0.732
Shell percentage (%)	8.92 ± 0.06 ^a^	7.85 ± 0.07 ^b^	8.89 ± 0.07 ^a^	7.90 ± 0.09 ^b^	0.959	**<0.001**	0.783
Elasticity (mm)	0.30 ± 0.06 ^a^	0.27 ± 0.04 ^b^	0.30 ± 0.04 ^a^	0.29 ± 0.05 ^a,b^	0.617	0.466	0.839
Breaking strength (g)	4394 ± 736 ^a^	3301 ± 693 ^b^	4296 ± 651 ^a^	3513 ± 867 ^b^	0.630	**<0.001**	0.191
Cuticle	0.34 ± 0.02	0.33 ± 0.03	0.30 ± 0.03	0.28 ± 0.04	0.082	0.531	0.991
Tibia weight (g)	10.94 ± 1.25 ^a,b^	11.63 ± 1.08 ^a^	10.7 ± 1.15 ^b^	11.38 ± 1.37 ^a,b^	0.326	**0.006**	0.999
Tibia diameter (mm)	7.27 ± 0.58	7.26 ± 0.45	7.15 ± 0.47	7.22 ± 0.52	0.493	0.829	0.654
Cortical Thickness (mm)	0.66 ± 0.09	0.61 ± 0.078	0.65 ± 0.09	0.63 ± 0.09	0.670	0.116	0.522
Tibia Breaking strength (g)	28,099 ± 4218 ^a,b^	27,086 ± 7317 ^a,b^	26,219 ± 5511 ^a^	30,989 ± 8142 ^b^	0.750	**0.040**	0.086
Tibia elasticity (mm)	2.24 ± 0.28	2.35 ± 0.40	2.32 ± 0.99	2.37 ± 0.42	0.736	0.479	0.821
Tibia Ashes (%)	35.21 ^a,b^ ± 2.89 ^a,b^	33.04 ^b^ ± 5.28 ^a^	34.50 ^b^ ± 4.47 ^a^	37.92 ^a^ ± 4.24 ^b^	**0.018**	0.471	**0.002**
PO4_AmideI CB	4.05 ± 0.36	4.08 ± 0.36	4.09 ± 0.43	4.17 ± 0.32	0.134	0.176	0.355
Mineral_CB (%)	57.17 ± 1.15	57.29 ± 0.97	57.63 ± 0.95	57.68 ± 0.80	**0.038**	0.682	0.865
PO4_AmideI MB	1.23 ± 0.40	1.22 ± 0.48	1.23 ± 0.47	1.36 ± 0.45	**0.081**	0.507	0.593
Mineral_MB (%)	38.43 ± 8.65	38.73 ± 7.37	36.73 ± 9.21	40.17 ± 6.30	0.936	0.265	0.347

**Table 3 animals-11-00623-t003:** Pearson’s correlation between egg related properties.

	Body Weight	Uterus Weight	Egg Weight	Egg White Height	Shell Weight	Shell Percentage	Shell Thickness	Shell Breaking Strength	Lay Percentage
Body weight	1.000								
Uterus weight	**0.343**	1.000							
Egg weight	**0.250**	**0.487**	1.000						
Egg white height	−0.119	−0.048	−0.011	1.000					
Shell weight	**0.254**	**0.352**	**0.503**	**−0.208**	1.000				
Shell percentage	0.115	0.082	−0.082	**−0.235**	**0.818**	1.000			
Shell thickness	0.127	0.191	0.137	**−0.236**	**0.876**	**0.919**	1.000		
Shell breaking strength	0.059	−0.011	−0.082	−0.185	**0.652**	**0.813**	**0.710**	1.000	
Lay percentage	**0.294**	**0.479**	0.043	−0.078	0.179	0.172	**0.205**	0.061	1.000

Significant values (*p* < 0.05) are in bold font.

**Table 4 animals-11-00623-t004:** Pearson’s correlation between some tibiae bone properties.

	Body Weight	Tibia Weight	Diameter	Cortical Thickness	Ash Percentage	Elasticity	Breaking Strength	Mineral%_CB	Mineral%_MB
Body weight	1.000								
Tibia weight	**0.601**	1.000							
Diameter	**0.216**	**0.265**	1.000						
Cortical thickness	**0.239**	0.080	**−0.221**	1.000					
Ash percentage	**0.255**	**0.211**	−0.171	**0.345**	1.000				
Elasticity	0.011	0.100	−0.186	0.042	**0.256**	1.000			
Breaking strength	**0.396**	**0.338**	−0.104	**0.327**	**0.751**	**0.376**	1.000		
% Mineral_CB	−0.073	−0.009	**−0.272**	0.043	0.145	−0.002	0.140	1.000	
% Mineral_MB	0.116	**0.256**	0.031	**0.216**	**0.656**	0.112	**0.454**	0.147	1.000

Significant values (*p* < 0.05) are in bold font.

**Table 5 animals-11-00623-t005:** Pearson’s correlation analysis between tibiae bone properties and egg properties.

		Egg Related Properties							
		Uterus Weight	Egg Weight	Egg White Height	Shell Weight	Shell Percentage	Shell Thickness	Shell Breaking Strength	Lay Percentage
**Bone related properties**	Tibia weight	0.194	0.055	−0.123	0.038	0.005	0.052	−0.081	0.197
	Diameter	0.156	**0.221**	−0.027	0.202	0.085	0.157	−0.039	0.076
	Cortical thickness	0.032	−0.102	−0.018	0.076	0.157	0.102	0.117	0.022
	Ash wet	0.020	−0.087	−0.113	0.011	0.077	0.052	0.102	−0.087
	Elasticity	0.052	0.003	−0.035	0.045	0.047	0.028	−0.022	−0.037
	Breaking strength	−0.034	−0.099	−0.065	−0.040	0.014	−0.009	0.024	0.039
	% Mineral_CB	**−0.237**	−0.173	0.040	−0.150	−0.053	−0.072	0.146	−0.109
	% Mineral_MB	−0.156	−0.087	−0.106	−0.107	−0.066	−0.088	−0.093	−0.117

Significant values (*p* < 0.05) are in bold font.

## Data Availability

Data used in this publication is available upon request to the corresponding authors.

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
