# Peer review of "Relationship between Bone Quality, Egg Production and Eggshell Quality in Laying Hens at the End of an Extended Production Cycle (105 Weeks)"

_animals, 2021, doi:10.3390/ani11030623_

Round 1
Reviewer 1 Report
This work is of great significance for understanding the relationship between layer bone quality and egg production performance. The aim of the experiment is relatively clear, and the materials & methods are clearly described. I recommend publishing this article after minor revisions.
- Please check the whole manuscript, a space should be added before and after “<”,”=”and ”>”。
- In Table 2, is it necessary to be accurate to "<0.0001"? As far as I know, it is sufficient when the p value is less than 0.001. Please explain this.
Author Response
We have made the recommended changes as described below:
- Please check the whole manuscript, a space should be added before and after “<”,”=”and ”>”。Answer: We have added the spaces as requested.
- In Table 2, is it necessary to be accurate to "<0.0001"? As far as I know, it is sufficient when the p value is less than 0.001. Please explain this. Answer: We agree with the reviewer and we have changed this value to 0.001
Reviewer 2 Report
Some minor changes will improve the readability of the manuscript:
Line 46: insert “in the” in front of “diet”
Line 47: change “start” to “starts”
Line 50: insert “the” in front of “night”
Line 53: change “specially” to “especially”
Line 63: insert “are” after “There”
Line 64: change “decreases” to “decrease”
Line 68: change “egg” to “eggs”
Line 70: change “makes” to “make”
Line 73: change “hens” to “hens’”
Line 75: change “On contrary” to “In contrast”
Line 100: delete “flock”
Line 105: In English, “fissured” would more likely be called “cracked”
Line 106: change “Two” to “two”
Line 154: change “weighted” to “weighed”
Line 159: give ATR in full first time used: attenuated total reflectance (ATR)
Line 167: change “calculate” to “calculated”
Line 182: change “being these values” to “these values being”
Line 230: change “show” to “shows”
Line 236: change “seems” to “seem”
Table 4: change “Thinckness” to “Thickness”
Line 326: delete “with”
Line 389: change “deteriorates” to “deteriorate”
Line 425, reference 3: Is this reference formatted correctly?
Author Response
We appreciated the seggestions and have done all corrections indicated by the referee:
Some minor changes will improve the readability of the manuscript:
Line 46: insert “in the” in front of “diet”
Line 47: change “start” to “starts”
Line 50: insert “the” in front of “night”
Line 53: change “specially” to “especially”
Line 63: insert “are” after “There”
Line 64: change “decreases” to “decrease”
Line 68: change “egg” to “eggs”
Line 70: change “makes” to “make”
Line 73: change “hens” to “hens’”
Line 75: change “On contrary” to “In contrast”
Line 100: delete “flock”
Line 105: In English, “fissured” would more likely be called “cracked”
Line 106: change “Two” to “two”
Line 154: change “weighted” to “weighed”
Line 159: give ATR in full first time used: attenuated total reflectance (ATR)
Line 167: change “calculate” to “calculated”
Line 182: change “being these values” to “these values being”
Line 230: change “show” to “shows”
Line 236: change “seems” to “seem”
Table 4: change “Thinckness” to “Thickness”
Line 326: delete “with”
Line 389: change “deteriorates” to “deteriorate”
Answer: We have done all suggested changes.
Line 425, reference 3: Is this reference formatted correctly? Answer: The reference format was not correct and has been modified.